# Advanced Pulmonary Sarcoidosis: Treatment and Monitoring

**DOI:** 10.3390/diagnostics15233061

**Published:** 2025-11-30

**Authors:** Gioele Castelli, Jessica Raja, Sam Banisadr, Vasileios Kouranos

**Affiliations:** 1Interstitial Lung Disease Unit, Royal Brompton Hospital Part of Guy’s and St Thomas’s Foundation Trust, London SW3 6NP, UK; gioelecastelli@gmail.com (G.C.); jessica.raja2@nhs.net (J.R.); drsbanisadr@gmail.com (S.B.); 2Respiratory Disease Unit, Department of Cardiac Thoracic, Vascular Sciences and Public Health, University of Padova, 35122 Padova, Italy

**Keywords:** pulmonary sarcoidosis, interstitial lung diseases, pulmonary hypertension, lung function tests, chest radiology, glucocorticoids, anti-inflammatory agents

## Abstract

Sarcoidosis is a multi-system inflammatory disease that poses several challenges to clinicians involved. Detecting advanced pulmonary sarcoidosis (APS) remains crucial at presentation as well as at every clinic visit since such a form of the disease is associated with an increased risk of developing respiratory failure, pulmonary hypertension (PH), and infectious complications, particularly in immunocompromised patients. As a result, APS accounts for the majority of sarcoidosis-related deaths. In this review, we focus on the definition of APS based on the identification of prognostic factors in pulmonary sarcoidosis. Additionally, we describe monitoring strategies for the detection of any disease progression and significant complications associated with the disease prognosis, such as pulmonary hypertension. Finally, the available treatment regimens for APS are reviewed with the aim of assessing the role of treatment strategies, especially since some patients are now eligible for antifibrotic treatment.

## 1. Introduction

Sarcoidosis is an inflammatory disease characterised histologically by the presence of non-caseating granulomatous inflammation in the involved organs. Despite years of research, the aetiology of sarcoidosis remains unclear. It has been hypothesised that the pathogenesis may involve both genetic predisposition and environmental factors [1]. One of the main challenges in the management of sarcoidosis is the heterogeneity of the disease at presentation and clinical course. Sarcoidosis can present at any age in both sexes, although the majority of patients are diagnosed between 30 and 50 years old [2]. A second peak between 40 and 60 years old has also been reported in women [3]. Different ethnicities have been found to have varied prevalence of disease. A significantly higher incidence has been observed in African American individuals compared with White Americans [4]. The clinical manifestations of the disease are variable, ranging from asymptomatic incidental findings with spontaneous remission to multi-system organ involvement [5,6]. Although a significant proportion of sarcoidosis patients will experience minimal or no symptoms with a favourable outcome, major organ involvement occurs in approximately 20–25% of patients that can lead to organ failure and permanent disability or even death [7]. The definition of advanced pulmonary sarcoidosis (APS) includes such forms of the disease at higher risk of progression to respiratory failure and death; this is predominantly patients with pulmonary fibrosis and pulmonary hypertension.

The lungs are the most commonly involved organ in sarcoidosis, and this can manifest with parenchymal and/or thoracic lymph abnormalities. While the mortality rate in sarcoidosis is estimated at 5–10%, death occurs predominantly in patients with advanced pulmonary disease. Therefore, detecting APS can provide both a risk stratification strategy and allow for optimal treatment planning to prevent adverse outcomes. Given the heterogenous disease course, treatment guidelines focus on (a) major organ involvement, aiming to prevent organ failure and disability, and/or (b) patients’ symptom management, aiming to improve their quality of life (QoL) [8,9]. Approximately 70% of patients with APS will require treatment, and a similar percentage will experience relapse after treatment withdrawal, suggestive of a more chronic disease course [10,11].

This review aims to evaluate the definition of APS in pulmonary sarcoidosis and identify factors associated with the development of advanced disease, particularly in patients with pulmonary fibrosis and pulmonary hypertension. Additionally, we will assess the currently available management strategies for APS, focusing on their effectiveness and related complications.

## 2. Advanced Pulmonary Sarcoidosis Definition

APS is a term used to describe forms of pulmonary disease associated with increased risk of permanent disability and mortality. Therefore, by definition, APS includes sarcoidosis patients with pulmonary fibrosis and sarcoidosis-associated pulmonary hypertension. Several studies have aimed to identify patients at increased risk of disease progression and mortality [12,13,14,15,16,17]. These studies have primarily focused on identifying baseline disease characteristics, many of which form the defining features of APS. More recently, an attempt has been made to evaluate the role of serial change during monitoring as a marker of disease progression and increased risk of adverse outcomes [18]. Table 1 summarises the key baseline characteristics associated with APS.

Presence and extent of pulmonary fibrosis on high-resolution computed tomography (HRCT) are well-established poor prognostic markers in pulmonary sarcoidosis. Furthermore, the integration of pulmonary function tests (PFT) and HRCT has been integrated to create a risk stratification tool as described by Walsh and colleagues. Walsh and colleagues created a prognostic score based on PFT and HRCT markers of severity [19]. Composite Physiological Index (CPI) was utilised to identify patients at the highest risk, using a threshold of 40. CPI is an index originally used in patients with idiopathic pulmonary fibrosis (IPF) to provide a validated marker of disease severity by adjusting for the effect of concomitant emphysema [20]. The cutoff of 40 seems to be equally effective in IPF and sarcoidosis in detecting patients with advanced disease [21]. Even when patients have less advanced disease in PFT (based on CPI), HRCT markers of severity appear to have a prognostic role in that study. An extent of fibrotic appearances greater than 20% of the lungs and/or an enlarged main pulmonary artery compared to the ascending aorta have been found to be an independent predictor of mortality and transplant in patients with pulmonary sarcoidosis [13,17,19]. In a large North American cohort, Kirkil et al. confirmed the value of Walsh staging. Age, detection of pulmonary hypertension on right heart catheterisation, and fibrotic extent over 20% on HRCT were associated with mortality in this cohort [13]. The Walsh staging system has also been validated in a French cohort, confirming the importance of integration of functional and radiological parameters in defining APS [17].

Pulmonary hypertension (PH) is an independent predictor of outcome in sarcoidosis, irrespective of the number and severity of organ involvement [17,22,23]. Sarcoidosis-associated PH (SAPH) can develop due to varied pathogenetic pathways, coexisting or isolated. These pathways include fibrotic lung disease, granulomatous compression of vessels, significant left heart disease, pulmonary veno-occlusive disease, as well as direct granulomatous inflammation of the pulmonary vessels [24]. SAPH increases the risk of developing respiratory failure and the need for supplemental oxygen, which can be viewed as a permanent disability [25]. SAPH is clearly part of the APS definition, as it may develop in patients without pulmonary fibrosis.

## 3. Pulmonary Fibrosis

Although approximately a quarter of pulmonary sarcoidosis patients are found to have pulmonary fibrosis on their baseline assessment, it is more common to develop fibrosis later in the disease course [12].

The mechanisms of fibrosis formation in pulmonary sarcoidosis remain poorly understood. Historically, fibrosis has been thought to develop as a result of untreated and progressive granulomatous inflammation [26]. The exact location of the granulomatous inflammation may lead to the formation of large conglomerates in the perihilar region, with peribronchovascular fibrosis associated with traction bronchiectasis and architectural distortion, and in some cases, the presence of honeycombing and a usual interstitial pneumonia (UIP) type pattern of fibrosis [27,28,29]. However, the development of UIP-type fibrosis as opposed to IPF remains an ongoing area of debate.

### 3.1. Clinical Presentation

The clinical presentation of pulmonary fibrosis can be highly variable, as is often the case in this heterogenous condition, with cough, shortness of breath, exertional dyspnoea, and chest tightness all reported. As these symptoms are present in a large number of chronic respiratory conditions, it is difficult to be particularly discriminatory. Auscultation may find fine crackles in a small group of patients [12]. However, this presentation, alongside digital clubbing, is rare in sarcoidosis patients with pulmonary fibrosis, especially when compared to patients with fibrotic interstitial lung disease.

Several blood biomarkers have been proposed for the diagnosis and monitoring of sarcoidosis [30]. The angiotensin-converting enzyme (ACE) has been used as a diagnostic tool, with elevated levels indicating active sarcoidosis. However, its sensitivity and specificity are not high enough to use it alone [30]. In pulmonary sarcoidosis, in particular, it does not have a value to predict prognosis or treatment response [31,32]. Soluble IL-2 receptor (sIL-2R) has shown more promise as a better prognostic tool [33]. However, it seems to be more applicable in extra-pulmonary sarcoidosis [34]. Other biomarkers, such as serum calcium and vitamin D levels, have been tested as prognostic tools without any success [30].

### 3.2. Pulmonary Function Tests

The lung function abnormalities in sarcoidosis pulmonary fibrosis can be varied; however, the most commonly described pattern is a restrictive ventilatory defect, with reduced total lung capacity (TLC). However, patients with APS may also develop obstructive patterns due to the type of architectural distortion. Mixed ventilatory defect (both restrictive and obstructive pattern) has been reported in approximately two-thirds of fibrotic pulmonary sarcoidosis patients [14]. The diffusing capacity for carbon monoxide (DLCO) is usually reduced. However, a disproportionately low DLCO in comparison to forced vital capacity (FVC) should raise suspicion for increased pulmonary artery pressures and pulmonary hypertension [35]. The presence of ventilatory defects has been related to higher mortality, especially when patients present with mixed ventilatory defects and DLCO reduction [14]. Kirkil et al. reported that a threshold of DLCO less than 40% predicted and FVC less than 50% predicted was associated with increased mortality in sarcoidosis patients in univariate analysis [13].

### 3.3. Imaging

Historically, CXR was used for the diagnosis of pulmonary fibrosis in sarcoidosis according to the Scadding staging system [36]. This staging system was developed based on differential presentations of mediastinal lymphadenopathy and pulmonary involvement. Stage I presented only hilar enlargement, associated with parenchymal involvement in Stage II. In Stage III, the CXR only presents non-fibrotic parenchymal involvement, while Stage IV is associated with pulmonary fibrosis. However, this classification has mild to poor agreement with lung function and symptomatic assessment of sarcoid patients [37,38]. Furthermore, poor interobserver agreement regarding the detection of stage IV disease has been reported even by experienced reporters [39]. Thus, HRCT remains the gold standard imaging tool to detect the presence and extent of pulmonary fibrosis [40]. An international Delphi consensus statement recently identified seven distinct HRCT patterns in pulmonary sarcoidosis associated with non-fibrotic and likely fibrotic presentations [41].

In APS, CXR findings are consistent with Scadding stage IV disease. Typical features include marked upper-lobe volume loss, retraction of the hila, coarse linear opacities, architectural distortion, traction bronchiectasis, and the presence of bullae or cyst-like lucencies. The changes are usually bilateral and most evident in the upper and central lung regions. Calcified lymph nodes can be seen in long-standing fibrosis [7,42,43].

Bronchocentric peribronchovascular fibrosis is the most common finding in fibrotic sarcoidosis. The fibrotic pattern on HRCT typically includes linear opacities most commonly radiating from the hila, causing architectural distortion in the upper lobes, and is often accompanied by traction bronchiectasis. The progressive massive fibrosis phenotype is a characteristic phenotype associated with significant airflow obstruction on PFT. The majority of fibrotic changes in pulmonary sarcoidosis are predominantly in the upper zones, and this can make calculating the extent of disease relative to the remaining lung more challenging. Fibrotic changes may occur in isolation or may coexist with inflammatory parenchymal changes such as micronodules and ground glass opacities. Cavity formation can occur in patients with fibrosis, and this often represents concomitant fungal infection. Additional features include mosaic attenuation due to small-airway involvement, pleural thickening, diaphragmatic tenting, and calcified mediastinal or hilar lymphadenopathy, collectively representing the end-stage sequelae of longstanding granulomatous inflammation [40,41]. Furthermore, a UIP pattern of fibrosis on HRCT can sometimes be seen in patients with sarcoidosis, creating a debate as to whether this represents a form of fibrotic pulmonary sarcoidosis or idiopathic pulmonary fibrosis [44].

18-fluorodeoxyglucose-positron emission tomography—computerised tomography (18-FDG-PET-CT or PET-CT) has been assessed as a tool to evaluate sarcoidosis activity. This imaging technique appears to have a high sensitivity and specificity for detecting both inflammatory activity in the lung and extra-thoracic organs [45]. Therefore, it can be used as a tool to guide immunosuppressive treatment in fibrotic disease. Interestingly, in patients with pulmonary sarcoidosis, 18-FDG uptake was present even in areas of fibrosis in 85% of subjects [46]. Furthermore, Adams et al. found no differences in FDG uptake between fibrotic and non-fibrotic sarcoidosis patients [47]. Therefore, areas of pulmonary fibrosis may contain ongoing active inflammation, suggesting that they may be susceptible to anti-inflammatory treatment. Thus, PET-CT may also be useful in guiding treatment strategies.

## 4. Monitoring of Advanced Pulmonary Sarcoidosis Patients

The monitoring of pulmonary sarcoidosis patients in relation to APS has 3 main aspects that are associated with different treatment strategies: (a) to evaluate evidence of disease progression indicating development of APS in patients with non-fibrotic pulmonary disease, (b) to assess any disease progression in patients with known APS suggestive of a progressive fibrotic phenotype, and (c) early detection of the development of pulmonary hypertension in patients with and without fibrotic pulmonary sarcoidosis. Figure 1 summarises the monitoring strategies for APS.

There are no guidelines regarding the frequency of monitoring in patients with and without APS. In clinical practice, this is usually individualised, but an assessment every 3–6 months is a sensible approach after APS detection. These intervals may be extended to 6–12 months in patients without APS. The time of treatment initiation as well as its intensity also influences the time of follow-up. A patient with a low risk of progressive disease who does not require treatment at baseline may be reviewed every 6–9 months in the first 2 years and then yearly if the condition remains stable. On the other hand, a patient initiated on induction treatment would benefit from a re-evaluation in around 2 months to evaluate the disease behaviour and the effect of high levels of immunosuppressive treatment [35].

Follow-up clinic review should include assessing (a) the burden of symptoms and the impact on QoL, (b) serial changes on PFT, which may prompt the performance of imaging, including HRCT and often PET-CT, (c) extra-pulmonary disease activity, and (d) presence or development of comorbidities affecting the patients’ symptoms and outcome. Certain questionnaires have been developed to evaluate the symptom burden in sarcoidosis patients, including the King’s Sarcoidosis Questionnaire [48] and the Fatigue Assessment Scale [49], but they have not been validated in patients with fibrotic pulmonary sarcoidosis. Using this approach allows respiratory physicians to accurately assess the patient’s baseline and identify patients with a higher risk of progressive pulmonary fibrosis [50]. However, each test in isolation has limited value due to different confounding factors.

The use of serial PFT is crucial in detecting progression of the parenchymal disease and/or development of pulmonary hypertension [51]. Depending on the ventilatory defect at presentation, different variables may be associated with disease progression in pulmonary sarcoidosis [51,52]. FVC is the most important parameter for monitoring disease and remains the gold standard endpoint in clinical trials, given its reproducibility and association with pulmonary fibrosis. DLCO is the most sensitive marker of inflammation and can also provide information regarding pulmonary vasculopathy. A significant decline in FVC associated with a concomitant decline in DLCO is more likely to indicate disease progression. While a relative change of >10% in FVC in 12 months from baseline would indicate definite disease progression, this is rarely observed in sarcoidosis. A decline of 5–10% in FVC values should prompt clinicians to review the serial DLCO change and further investigate with chest imaging [53]. A similar approach should be considered in the event of DLCO decline > 15% in 12 months from baseline, but the development of pulmonary vasculopathy should also be excluded in this situation. Due to concerns with variability in testing, we would recommend reviewing changes in both FVC and DLCO as well as integrating any respiratory symptom changes before making further decisions about management changes.

Although CXR has been recommended as an indicator of disease progression whenever there are changes in symptoms and/or PFT trends [37], serial CXR has been found to have limited correlation with disease progression [39]. On the other hand, HRCT can be even more sensitive than PFT. Gafá et al. assessed 14 consecutive scans of pulmonary sarcoidosis patients and demonstrated that 58% had radiological progression with only a partial correlation with FVC change [54]. Furthermore, HRCT could provide insight regarding complications associated with changes in symptoms and/or PFT, including detection of pulmonary embolism, mycetoma formation, or other infectious complications [55]. An individualised approach to radiological monitoring should be considered for each patient at every clinic visit, depending on the risk stratification, treatment goal, and QoL issues [35]. The 18FDG-PET-CT has now become an important monitoring tool in sarcoidosis [45]. Change in FDG-PET scans has been used for detecting disease progression due to ongoing inflammation and has been strongly correlated with a significant decline in DLCO [56]. Nonetheless, its role in the monitoring of APS has not yet been established or validated.

## 5. Sarcoidosis Associated Pulmonary Hypertension (SAPH)

The incidence of PH in sarcoidosis patients is estimated at 5–10% of the general population but is expected to increase to >50% in symptomatic patients with APS. The development of SAPH is multifactorial, which has led to its inclusion in the miscellaneous group of the World Health Organisation (WHO) classification of PH subtypes [57]. The main pathogenetic mechanisms of SAPH are shown in Table 2. Several surrogate non-invasive markers are used to screen patients for the development of SAPH and are summarised in Table 3 [58]. Echocardiography remains the gold standard tool for identifying patients who should undergo right heart catheterisation (RHC) according to the guidelines [57]. Nonetheless, a combination of the surrogate markers of PH, even in the absence of echocardiographic findings suggestive of PH, should lead to discussions between clinicians and PH experts for the performance of RHC. PH is now defined by a mean pulmonary arterial pressure (mPAP) of greater than 20 mmHg measured during RHC studies and can be further categorised into pre-capillary and post-capillary depending on the pulmonary arterial wedge pressure and elevated pulmonary vascular resistance [57].

## 6. Treatment Strategies in Sarcoidosis

APS represents a form of major organ involvement in sarcoidosis, and therefore, treatment may be indicated or already initiated in the majority of patients. One of the key principles of treatment lies in identifying patients at high risk of progression. Immunosuppressive treatment is considered to prevent further progression of pulmonary fibrosis by eliminating inflammatory activity and remains the first treatment choice. Nonetheless, APS patients are usually excluded from randomised controlled trials investigating the effectiveness of immunosuppression. Treatment guidelines developed for the general sarcoidosis population are also used for APS patients [8]. Furthermore, recent studies have detected sarcoidosis patients with progressive pulmonary fibrosis despite conventional immunosuppression who responded well to the introduction of antifibrotic treatment [59,60]. The role of immunosuppression in the management of pulmonary vasculopathy remains unclear. Finally, early referral for lung transplantation should be considered in all APS patients with evidence of progression.

Corticosteroids are considered the first drug of choice even in APS. Nonetheless, steroid-sparing agents should be introduced early if there is evidence of active pulmonary inflammation or inability to reduce the prednisolone dose. The recently published SARCORT trial found that a starting dosage of 40 mg does not differ from an initial dosage of 20 mg when evaluating treatment failure and sarcoidosis relapse as endpoints [61]. APS is usually associated with chronic disease requiring prolonged treatment. A strategy to gradually withdraw steroid treatment with early introduction of steroid-sparing agents should be strongly considered when long-term treatment is required. Long-term treatment with steroids should be avoided, as this often leads to significant side effects, including obesity, diabetes, arterial hypertension, osteoporosis, ocular diseases, and mood and sleep disturbances that are associated with the treatment duration and the cumulative dosage exposure [62,63,64]. Therefore, the use of steroids, especially in the context of severe organ damage such as APS, may not be avoided. It should, in any case, be associated with the use of steroid-sparing agents to reduce the exposure to 3–6 months [65].

Methotrexate is the first-choice steroid-sparing agent in high-risk sarcoidosis [8], as shown in observational studies, both prospective and retrospective [66,67,68]. Randomised controlled trials have been designed to assess the role of methotrexate as a first-line agent in pulmonary sarcoidosis, but APS has not been included [69]. Nonetheless, it seems logical to use methotrexate monotherapy in patients with fibrotic sarcoidosis when treatment is indicated [70]. Treatment with methotrexate may be stopped after two years; however, in 80% of cases, the disease relapses after withdrawal [71]. Other second-line drugs, such as azathioprine, leflunomide, and mycophenolate, showed efficacy in retrospective observational studies and were included in the most recent treatment guidelines [8,68,72,73,74]. Biological agents targeting the tumour necrosis factor (TNF) pathway, such as infliximab and adalimumab, have been approved for use in refractory sarcoidosis, but their role in APS has not yet been clarified [75]. In pulmonary sarcoidosis, infliximab treatment did not result in significant FVC change [76], but its role in other major organ involvement, such as neurosarcoidosis and cardiac sarcoidosis, is now considered to be established [77,78]. Small retrospective studies showed promising results from Janus kinase (JAK), such as ruxolitinib, and anti-interleukine-6 agents, such as tocilizumab [79,80]. However, these options have not been approved for wider use and should be proposed on a case-by-case basis. Finally, efzofitimod, an intravenous neuropilin-2 inhibitor with immunomodulatory effects, is currently in a phase III clinical trial (NCT05415137) with promising results in earlier stages [81].

The usage of immunomodulators and immunosuppressants makes sarcoidosis patients more susceptible to infections, particularly with higher steroid doses and immunosuppressants [82]. Methotrexate appears to be associated with fewer infections in a retrospective observational study comparing methotrexate with azathioprine as second-line agents [68]. In APS patients, there is a higher risk of developing chronic pulmonary aspergillosis given the intensity of immunosuppression and the presence of fibrotic disease. Typically, this appears as mycetomas within fibrocavitary disease, with typical symptoms including weight loss and haemoptysis [55]. Chronic pulmonary aspergillosis has been associated with a worse outcome in patients with APS and is often associated with the development of concomitant SAPH.

When the immunosuppressive regimen fails to control the progression of the disease, a progressive pulmonary fibrosis phenotype has been described in fibrotic interstitial lung diseases other than idiopathic pulmonary fibrosis, including fibrotic pulmonary sarcoidosis. Nintedanib, a small molecule inhibiting different growth factors, has been approved for use in these populations to slow disease progression following the results of the INBUILD trial [59]. Notably, the group of sarcoidosis patients included in the INBUILD trial was small, and further studies are warranted to evaluate the role of antifibrotics in the sarcoidosis population [60]. The use of pirfenidone, the other antifibrotic agent approved for IPF, has not been confirmed in other fibrotic and progressive diseases. In particular, the RELIEF trial focused on progressive fibrosis, but despite promising results, it was prematurely discontinued due to slow recruitment. Notably, it did not include any patients with APS [83].

In the presence of SAPH, several vasodilating drugs have been tested or are used off-label, with conflicting results [84]. Although PH-specific medications may not improve outcomes, they have been associated with improved symptoms (dyspnoea score and 6-min walking test distance). Bosentan, an endothelin receptor antagonist, showed an improvement in hemodynamics of SAPH patients but did not reach any clinical outcome [85]. Iloprost, a prostacyclin analogue, in its inhaled preparation, showed improvement in quality of life and clinical improvement in a subgroup of SAPH patients [86]. Riociguat, a soluble guanylate cyclase stimulator, showed efficacy in prolonging the time to clinical worsening [87].

## 7. Conclusions

APS accounts for the majority of transplant referrals and sarcoidosis-related deaths. The diagnosis of this phenotype of sarcoidosis requires clinical, functional, and radiological evaluations. This initial evaluation helps in the prognostic stratification of APS patients. However, monitoring of disease activity is crucial in determining treatment decisions to reduce the risk of progression. Treatment strategies include steroids and immunosuppressive drugs, and in APS, an aggressive “top-down” approach may be required.


## Figures and Tables

**Figure 1 diagnostics-15-03061-f001:**
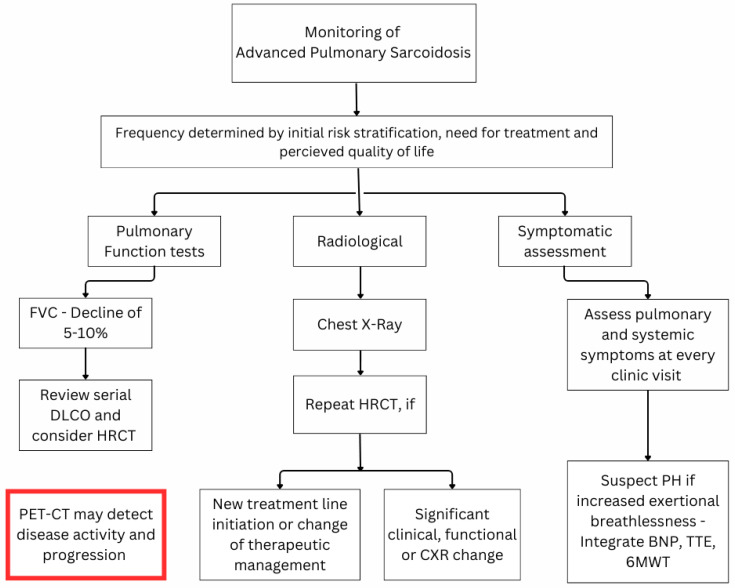
Proposed monitoring algorithm for advanced pulmonary sarcoidosis. Abbreviations: 6MWT = six-minute walking test; BNP = brain natriuretic peptide; DLCO = lung diffusion of carbon monoxide; FVC = forced vital capacity; HRCT = high-resolution computed tomography; PET-CT = positron emission tomography—computed tomography; PH = pulmonary hypertension; TTE = trans-thoracic echocardiogram.

**Table 1 diagnostics-15-03061-t001:** Key characteristics of advanced pulmonary sarcoidosis.

Key Characteristics of Advanced Pulmonary Sarcoidosis		
Diagnostic tools		
PFTs	Composite Physiological Index (CPI)	>40
	Diffusing capacity of carbon monoxide (DLCO)	<40–60% predicted
	Forced vital capacity (FVC)	<50–70% predicted
HRCT	Extent of pulmonary fibrosis	>15–20%
	Main pulmonary artery to ascending aorta ratio	>1
	Right ventricle/left ventricle ratio	>1
Echocardiography	Reduced right ventricular function	
	TAPSE	<20 mm
	Right ventricular outflow acceleration time	<105 m/s
	Increased right ventricular size	
	Elevated estimated pulmonary artery systolic pressure	>40 mmHg
	Flattening of the interventricular septum	
	Tricuspid regurgitant velocity	>2.8 m/s
RHC	Mean pulmonary artery pressure	>20 mmHg

Abbreviations: PFTs: Pulmonary function tests, HRCT: High-resolution computer tomography, RHC: Right heart catheterisation.

**Table 2 diagnostics-15-03061-t002:** Main pathogenetic mechanism of sarcoidosis-associated pulmonary hypertension (SAPH).

1. Advanced pulmonary sarcoidosis
2. Left heart disease and possibly cardiac sarcoidosis
3. Pulmonary veno-occlusive disease
4. Thromboembolic disease
5. Pulmonary vascular compression from fibrotic strictures (fibrotic mediastinitis) and/or lymph nodes
6. Direct vascular granulomatous inflammation
7. Comorbidities including anaemia and liver disease

**Table 3 diagnostics-15-03061-t003:** Surrogate markers of sarcoidosis-associated pulmonary hypertension.

Surrogate Markers of PH	
PFT	DLCO < 50%
	Kco < 60%
	FVC/DLCO > 1.6
	Serial DLCO decline > 15% in 12 months with or without FVC change
HRCT	Fibrosis extent > 20%
	MPA/AA > 1
	Right/left ventricle ratio > 1
Echocardiography	Tricuspid regurgitant velocity > 2.8 m/s
	Right/left ventricle ratio > 1
	Flattening of the interventricular septum
	Right ventricular outflow acceleration < 105 m/s
	Early diastolic pulmonary regurgitation > 2.2 m/s
	Pulmonary artery, inferior vena cava, or right atrial dilatation
6MWT	<350 m
	Saturation drop > 5%
	Serial drop in 6MWD > 20% without FVC change
BNP	>100 ng/lt
A/a gradient	Widening (>10 mmHg)
Clinical	Worsening of NYHA class

Abbreviations: 6MWT = six-minute walking test; A/a = Arterial/alveolar; BNP = brain natriuretic peptide; DLCO = lung diffusion of carbon monoxide; FVC = forced vital capacity; HRCT = high-resolution computed tomography; Kco = carbon monoxide transfer coefficient; MPA/AA = main pulmonary artery to ascending aorta ratio; PFT = pulmonary function tests; PH = pulmonary hypertension.

## Data Availability

No new data were created or analyzed in this study.

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
