# Peer review of "Advanced Pulmonary Sarcoidosis: Treatment and Monitoring"

_diagnostics, 2025, doi:10.3390/diagnostics15233061_

Round 1
Reviewer 1 Report
Comments and Suggestions for Authors
The article is a useful review of the currently available diagnostic tools for the clinical management of patients with sarcoidosis and advanced lung disease. The main question addressed by the authors is to identify patients with advanced pulmonary sarcoidosis (APC). The topic is relevant because it may be useful to clinicians in identifying patients who require early treatment and close follow-up.
The authors' main contribution to the field is to summarize in a didactic way the prognostic factors of APC, classifying them according to clinical, functional, and radiological criteria. Although the review carried out by the authors does not add new information, it adequately summarizes the existing data in the field. Also, the references are extensive and up-to-date. There are some self-citations, but they are justified based on the extensive experience of the authors in the field.
Regarding antifibrotic drugs, no mention is made of the potential role of pirfenidone in patients with progressive pulmonary fibrosis.
The potential benefit of pulmonary vasodilators in patients with significant pulmonary hypertension is not mentioned.
Author Response
We appreciate the positive comments from the reviewer regarding our work.
- Regarding antifibrotic drugs, no mention is made of the potential role of pirfenidone in patients with progressive pulmonary fibrosis.
We thank the reviewer for the suggestion. We added a comment on pirfenidone in the end of the antifibrotic dedicated paragraph. - The potential benefit of pulmonary vasodilators in patients with significant pulmonary hypertension is not mentioned.
We thank the reviewer for the comment. We added a separate paragraph to discuss briefly the literature regarding vasodilators in SAPH.
Reviewer 2 Report
Comments and Suggestions for Authors
This is a review article on the monitoring and treatment of advanced pulmonary sarcoidosis (APS).
This review describes the definition of APS, monitoring strategies for disease progression, and treatment strategies. This is a well written and detailed manuscript. This review article provides some useful information to readers. I have a few comments.
- Please add the chest X-ray and chest CT features of advanced pulmonary sarcoidosis.
- In line 139, I think it would be better to describe the stage system of Sarcoidosis (from stage I to stage IV).
- There was no description of laboratory finding of APS. I think it would be better to describe the laboratory finding of APS. (ex, increased level of serum calcium, ACE etc)
- There was no treatment period of APS in the “Treatment strategies in sarcoidosis” section. Please describe the treatment period of steroid and MTX.
The English could be improved to more clearly express the research.
Author Response
We appreciate the positive comments from the reviewer regarding our work.
- Please add the chest X-ray and chest CT features of advanced pulmonary sarcoidosis.
We expanded our presentation of the CXR and CT features of APS in the section dedicated to diagnosis. - In line 139, I think it would be better to describe the stage system of Sarcoidosis (from stage I to stage IV).
We thank the reviewer for the suggestion. We added a brief description of Stage I to IV in the text.
3. There was no description of laboratory finding of APS. I think it would be better to describe the laboratory finding of APS. (ex, increased level of serum calcium, ACE etc)
A new paragraph has been added in the clinical evaluation section to answer this comment. Nonetheless, we would like to note that there are no laboratory findings of APS
4.There was no treatment period of APS in the “Treatment strategies in sarcoidosis” section. Please describe the treatment period of steroid and MTX.
We thank the reviewer for the comment. We added a recent position paper on the importance of rapid steroid withdrawal and introduction of steroid sparing agents. We also added the timing for MTX stop.
The English could be improved to more clearly express the research.
We find the comment inappropriate and largely offensive. If any concerns the comment needs to be more specific. The manuscript has been written and edited in detail.
We would like to confirm that the Tables and figures given in the manuscript are all original work.